# Finding Friend and Foe in Multi-Agent Games

**Jack Serrino**[*]
MIT
jserrino@mit.edu

**Max Kleiman-Weiner**[*]
Harvard, MIT, Diffeo
maxkleimanweiner@fas.harvard.edu

**David C. Parkes**
Harvard University
parkes@eecs.harvard.edu

**Joshua B. Tenenbaum**
MIT, CBMM
jbt@mit.edu

## Abstract

Recent breakthroughs in AI for multi-agent games like Go, Poker, and Dota, have seen great strides in recent years. Yet none of these games address the real-life challenge of cooperation in the presence of unknown and uncertain teammates. This challenge is a key game mechanism in hidden role games. Here we develop the DeepRole algorithm, a multi-agent reinforcement learning agent that we test on *The Resistance: Avalon*, the most popular hidden role game. DeepRole combines counterfactual regret minimization (CFR) with deep value networks trained through self-play. Our algorithm integrates deductive reasoning into vector-form CFR to reason about joint beliefs and deduce partially observable actions. We augment deep value networks with constraints that yield interpretable representations of win probabilities. These innovations enable DeepRole to scale to the full Avalon game. Empirical game-theoretic methods show that DeepRole outperforms other hand-crafted and learned agents in five-player Avalon. DeepRole played with and against human players on the web in hybrid human-agent teams. We find that DeepRole outperforms human players as both a cooperator and a competitor.

## 1   Introduction

Cooperation enables agents to achieve feats together that no individual can achieve on her own [16, 39]. Cooperation is challenging, however, because it is embedded within a competitive world [15]. Many multi-party interactions start off by asking: who is on my team? Who will collaborate with me and who do I need to watch out for? These questions arise whether it is your first day of kindergarten or your first day at the stock exchange. Figuring out who to cooperate with and who to protect oneself against is a fundamental challenge for any agent in a diverse multi-agent world. This has been explored in cognitive science, economics, and computer science [2, 7, 8, 21, 23, 24, 25, 26, 28, 30, 31, 44].

Core to this challenge is that information about who to cooperate with is often noisy and ambiguous. Typically, we only get this information indirectly through others' actions [1, 3, 21, 41]. Since different agents may act in different ways, these inferences must be robust and take into account ad-hoc factors that arise in an interaction. Furthermore, these inferences might be carried out in the presence of a sophisticated adversary with superior knowledge and the intention to deceive. These adversaries could intentionally hide their non-cooperative intentions and try to appear cooperative for their own benefit [36]. The presence of adversaries makes communication challenging— when intent to cooperate is unknown, simple communication is unreliable or "cheap" [14].

This challenge has not been addressed by recent work in multi-agent reinforcement learning (RL). In particular, the impressive results in imperfect-information two-player zero-sum games such as poker

---

[*]indicates equal contribution

[4, 6, 27] are not straightforward to apply to problems where cooperation is ambiguous. In heads-up poker, there is no opportunity to actually coordinate or cooperate with others since two-player zero-sum games are strictly adversarial. In contrast, games such as Dota and capture the flag have been used to train Deep RL agents that coordinate with each other to compete against other teams [17, 29]. However, in neither setting was there ambiguity about *who* to cooperate with. Further in real-time games, rapid reflexes and reaction times give an inherent non-strategic advantage to machines [9].

Here we develop *DeepRole*, a multi-agent reinforcement learning algorithm that addresses the challenge of learning who to cooperate with and how. We apply DeepRole to a five-player game of alliances, *The Resistance: Avalon* (Avalon), a popular hidden role game where the challenge of learning *who* to cooperate with is the central focus of play [13]. Hidden role games start with players joining particular teams and adopting roles that are not known to all players of the game. During the course of the game, the players try to infer and deduce the roles of their peers while others simultaneously try to prevent their role from being discovered. As of May 2019, Avalon is the most highly rated hidden role game on boardgamegeek.com. Hidden role games such as Mafia, Werewolf, and Saboteur are widely played around the world.

**Related work**  DeepRole builds on the recent success of heuristic search techniques that combine efficient depth-limited lookahead planning with a value function learned through self-play in two-player zero-sum games [27, 33, 34]. In particular, the DeepStack algorithm for no-limit heads up poker combines counterfactual regret minimization (CFR) using a continual re-solving local search strategy with deep neural networks [27, 45]. While DeepStack was developed for games where all actions are public (such as poker), in hidden role games some actions are only observable by some agents and therefore must be deduced. In Avalon, players obtain new private information as the game progresses while in poker the only hidden information is the initial set of cards.

**Contributions.**  Our key contributions build on these recent successes. Our algorithm integrates deductive reasoning into vector-form CFR [19] to reason about joint beliefs and partially observable actions based on consistency with observed outcomes, and augments value networks with constraints that yield interpretable representations of win probabilities. This augmented network enables training with better sample efficiency and generalization. We conduct an empirical game-theoretic analysis in five-player Avalon and show that the DeepRole CFR-based algorithm outperforms existing approaches and hand-crafted systems. Finally, we had DeepRole play with a large sample of human players on a popular online Avalon site. DeepRole outperforms people as both a teammate and opponent when playing with and against humans, even though it was only trained through self-play. We conclude by discussing the value of hidden role games as a long-term challenge for multi-agent RL systems.

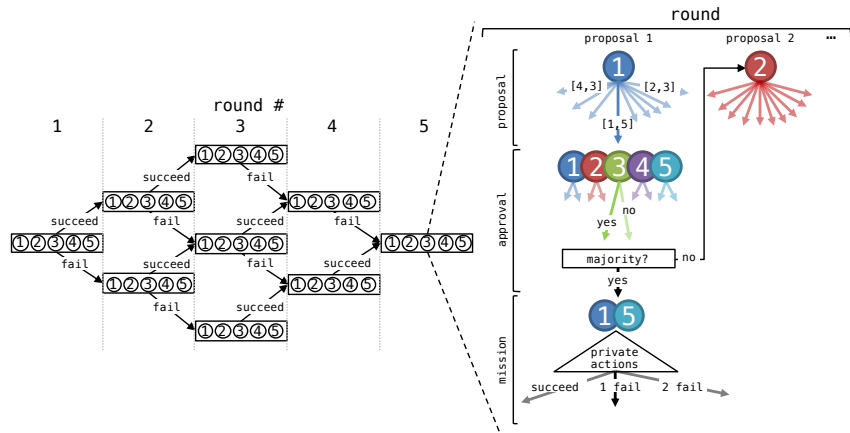

Figure 1: Description of the public game dynamics in The Resistance: Avalon. (left) Each round (rectangle) has up to 5 proposals (white circles) and leads to either a mission that fails or succeeds. (right) Example dynamics within each round. Players (colored circles) alternate proposing subsets of players (2 or 3) to go on a mission which are then put to vote by all 5 players. If the majority approves, those players (1 & 5 in this example) privately and independently decide to succeed or fail the mission. If the majority disapproves, the next player proposes a subset.

## 2 The Resistance: Avalon

We first briefly describe game mechanics of The Resistance: Avalon played with five players. At the beginning of the game, 3 players are randomly assigned to the *Resistance* team and 2 players are assigned to the *Spy* team. The spies know which players are on the Spy team (and hence also know which players are on the Resistance team). One member of the Resistance team is randomly and privately chosen to be the *Merlin* role who also knows all role assignments. One member of the Spy team is randomly chosen to be the *Assassin*. At the end of the game, if the Resistance team has won, the Assassin guesses the identity of Merlin. If the Assassin guesses Merlin correctly then the Spy team wins.

Figure 1 shows a visual description of the public game dynamics. There are five *rounds* in the game. During each round a player *proposes* a subset (two or three depending on the round) of agents to go on a *mission*. All players simultaneously and publicly vote (approve or not approve) of that subset. If a simple majority do not approve, another player is selected to propose a subset to go on the mission. If after five attempts, no proposal receives a simple majority, the Spy team wins. If a simple majority approves, the subset of players privately select whether the mission succeeds or fails. Players on the Resistance team must always choose success but players on the Spy team can choose success or failure. If any of the Spies choose to fail the mission, the mission fails. Otherwise, the mission succeeds. The total number of success and fail votes is made public but the identity of who made those votes is private. If three missions succeed, the Resistance team wins. If three missions fail, the Spy team wins. When people play Avalon, the games are usually rich in "cheap talk," such as defending oneself, accusing others, or debunking others' claims [10]. In this work, we do not consider the strategic implications of natural language communication.

Although Avalon is a simple game to describe, it has a large state space. We compute a lower bound of $10^{56}$ distinct information sets in the 5-player version of Avalon (Appendix D for details). This is larger than the state space of Chess ($10^{47}$) and larger than the number of information sets in heads-up limit poker ($10^{14}$) [18].

## 3 Algorithm: DeepRole

The DeepRole algorithm builds off of recent success in poker by combining DeepStack's innovations of deep value networks and depth-limited solving with deductive reasoning. Compared to MCTS-based methods like AlphaGo, CFR-based methods like DeepStack and DeepRole can soundly reason over hidden information. Unique to DeepRole, our innovations allow the algorithm to play games with simultaneous and hidden actions. In broad strokes, DeepRole is composed of two parts: (1) a CFR planning algorithm augmented with deductive reasoning; and (2) neural value networks that are used to reduce the size of the game tree. Source code and experimental data is available here: https://github.com/Detry322/DeepRole.

**Background.** Hidden role games like Avalon can be modeled as extensive-form games. We follow the notation of [19]. Briefly, these games have a game tree with nodes that correspond to different histories of actions, $h \in H$, with $Z \subset H$ the set of terminal histories. For each $h \in Z$, let $u_i(h)$ be the utility to player $i$ in terminal history $h$. In extensive-form games, only a single player $P(h)$ can move at any history $h$, but because Avalon's mechanics intimately involve simultaneous action, we extend this definition to let $P'(h)$ be the set of players simultaneously moving at $h$. Histories are partitioned into information sets ($I \in \mathcal{I}_i$) that represent the game states that player $i$ cannot distinguish between. For example, a Resistance player does not know who is on the Spy team and thus all $h$ differing only in the role assignments to the other players are in a single information set. The actions available in a given information set are $a \in A(I)$.

A strategy $\sigma_i$ for player $i$ is a mapping for each $I \in \mathcal{I}_i$ to a probability distribution over $A(I)$. Let $\sigma = (\sigma_1, \ldots, \sigma_p)$ be the joint strategy of all $p$ players. We write $\sigma_{I \to a}$ to mean strategy $\sigma$, modified so action $a$ is always played at information set $I$. Then, we let $\pi^\sigma(h)$ be the probability of reaching $h$ if all players act according to $\sigma$. We write $\pi_i^\sigma(h)$ to mean the contribution of player $i$ to the joint probability $\pi^\sigma(h) = \prod_{1 \ldots p} \pi_i^\sigma(h)$. Finally, let $\pi_{-i}^\sigma(h)$ be the product of strategies for all players except $i$ and let $\pi^\sigma(h, h')$ be the probability of reaching history $h'$ under strategy $\sigma$, given $h$ has occurred.

Counterfactual regret minimization (CFR) iteratively refines $\sigma$ based on the regret accumulated through a self-play like procedure. Specifically, in CFR+, at iteration $T$, the cumulative counterfactual regret is $R_i^{+,T}(I,a) = \max\{\sum_T CFV_i(\sigma_{I \to a}^t, I) - CFV_i(\sigma^t, I), 0\}$ where the counterfactual values for player $i$ are defined as $CFV_i(\sigma, I) = \sum_{z \in Z} u_i(z)\pi_{-i}^\sigma(z[I])\pi^\sigma(z[I], z)$, where $z[I]$ is the $h \in I$ such that $h \sqsubseteq z$ [38]. At a high-level, CFR iteratively improves $\sigma$ by boosting the probability of actions that would have been beneficial to each player. In two-player zero-sum games, CFR provably converges to a Nash equilibrium. However, it does not necessarily converge to an equilibrium in games with more than two players [37]. We investigate whether CFR can generate strong strategies in a multi-agent hidden role game like Avalon.

### 3.1 CFR with deductive logic

The CFR component of DeepRole is based on the vector-form public chance sampling (PCS) version of CFR introduced in [19], together with CFR+ regret matching [38]. Vector-form versions of CFR can result in faster convergence and take advantage of SIMD instructions, but require a public game tree [20]. In poker-like games, one can construct a public game tree from player actions, since all actions are public (e.g., bets, new cards revealed) except for the initial chance action (giving players cards). In hidden role games, however, key actions after the initial chance action are made privately, breaking the standard construction.

To support hidden role games, we extend the public game tree to be a history of third-person observations, $o \in O(h)$, instead of just actions. This includes both public actions and observable consequences of private actions (lines 22-44 in Alg. 1 in the Appendix). Our extension works when deductive reasoning from these observations reveals the underlying private actions. For instance, if a mission fails and one of the players is known to be a Spy, one can deduce that the Spy failed the mission. deduceActions$(h, o)$ carries out this deductive reasoning and returns the actions taken by each player under each information set ($\vec{a}_i[I]$) (line 23). With $\vec{a}_i[I]$ and the player's strategy ($\vec{\sigma}_i$), the player's reach probabilities are updated for the public game state following the observation ($ho$) (lines 24-26).

Using the public game tree, we maintain a human-interpretable joint posterior belief $\mathbf{b}(\rho|h)$ over the initial assignment of roles $\rho$. $\rho$ represents a full assignment of roles to players (the result of the initial chance action) – so our belief $\mathbf{b}(\rho|h)$ represents the joint probability that each player has the role specified in $\rho$, given the observed actions in the public game tree. See Figure 2 for an example belief $\mathbf{b}$ and assignment $\rho$. This joint posterior $\mathbf{b}(\rho|h)$ can be approximated by using the individual players' strategies as the likelihood in Bayes rule:

$$\mathbf{b}(\rho|h) \propto \mathbf{b}(\rho)(1 - \mathbb{1}\{h \vdash \neg\rho\}) \prod_{i \in 1...p} \pi_i^\sigma(I_i(h, \rho)) \tag{1}$$

where $\mathbf{b}(\rho)$ is the prior over assignments (uniform over the 60 possible assignments), $I_i(h, \rho)$ is the information set implied by public history $h$ and assignment $\rho$, and the product is the likelihood of playing to $h$ given each player's implied information set. A problem is that this likelihood can put positive mass on assignments that are impossible given the history. This arises because vector-form CFR algorithms can only compute likelihoods for each player independently rather than jointly. For instance, consider two players that went on a failing mission. In the information sets implied by the $\rho$ where they are both resistance, each player is assumed to have passed the mission. However, this is logically inconsistent with the history, as one of them must have played fail. To address this, the indicator term $(1 - \mathbb{1}\{h \vdash \neg\rho\})$ zeros the probability of any $\rho$ that is logically inconsistent with the public game tree $h$. This zeroing removes any impact these impossible outcomes would have had on the value and regret calculations in CFR (line 20 in Alg. 2).

### 3.2 Value network

The enhanced vector-form CFR cannot be run on the full public game tree of Avalon (or any real hidden role game). This is also the case for games like poker, so CFR-based poker systems [6, 27] rely on action abstraction and state abstraction to reduce the size of the game tree. However, actions in Avalon are not obviously related to each other. Betting 105 chips in poker is strategically similar to betting 104 chips, but voting up a mission in Avalon is distinct from voting it down. The size of Avalon's game tree does not come from the number of available actions, but rather from the number of players. Furthermore, since until now Avalon has only received limited attention, there are no

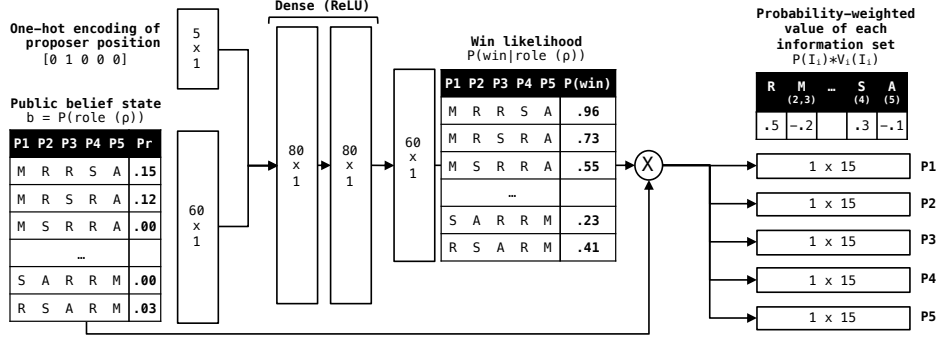

Figure 2: DeepRole neural network architecture used to limit game tree depth. Tables (black headers) show example inputs. The uppercase characters represent the different roles: (R)esistance, (S)py, (M)erlin, (A)ssassin. The outputs are the probability weighted value for each player in each of their information sets. While there is only one information set for Resistance (since they only know their own role), there are multiple for each of the other roles types. "M (2,3)" should be read as Merlin who sees players 2 and 3 as Spy and "S (4)" should be read as Spy who sees player 4 as Assassin.

developed hand-crafted state abstractions available either (although see [5] for how these could be learned). We follow the general approach taken by [27], using deep neural networks to limit the size of the game tree that we traverse (lines 14-16 in Alg. 1 in Appendix A).

We first partition the Avalon public game tree into individually solvable parts, segmented by a proposal for every possible number of succeeded and failed missions (white circles on the left side of Figure 1). This yields 45 neural networks. Each $h$ corresponding to a proposal is mapped to one of these 45 networks. These networks take in a tuple $\theta \in \Theta, \theta = (i, \mathbf{b})$ where $i$ is the proposing player, and $\mathbf{b}$ is the posterior belief at that position in the game tree. $\Theta$ is the set of all possible game situations. The value networks are trained to predict the probability-weighted value of each information set (Figure 2).

Unlike in DeepStack, our networks calculate the *non-counterfactual* (i.e. normal) values for every information set $I$ for each player. This is because our joint belief representation loses the individual contribution of each player's likelihood, making it impossible to calculate a counterfactual. The value $V_i(I)$ for private information $I$ for player $i$ can be written as:

$$V_i(I) = \pi_i^\sigma(I) \sum_{h \in I} \pi_{-i}^\sigma(h) \sum_{z \in Z} \pi^\sigma(h, z) u_i(z) = \pi_i^\sigma(I) \, CFV_i(I)$$

where players play according to a strategy $\sigma$. Since we maintain a $\pi_i^\sigma(I)$ during planning, we can convert the values produced by the network to the counterfactual values needed by CFR (line 15 in Alg. 2).

**Value network architecture**   While it's possible to estimate these values using a generic feed-forward architecture, it may cause lower sample efficiency, require longer training time, or fail to achieve a low loss. We design an interpretable custom neural network architecture that takes advantage of restrictions imposed by the structure of many hidden role games. Our network feeds a one-hot encoded vector of the proposer player $i$ and the belief vector $\mathbf{b}$ into two fully-connected hidden layers of 80 ReLU units. These feed into a fully-connected *win probability layer* with sigmoid

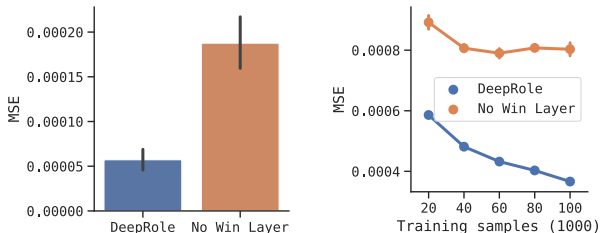

Figure 3: DeepRole generalization and sample efficiency. (left) Generalization error on held out samples averaged across the 45 neural networks. (right) Generalization error as a function of training data for the first deep value network (averaged over N=5 runs, intervals are SE).

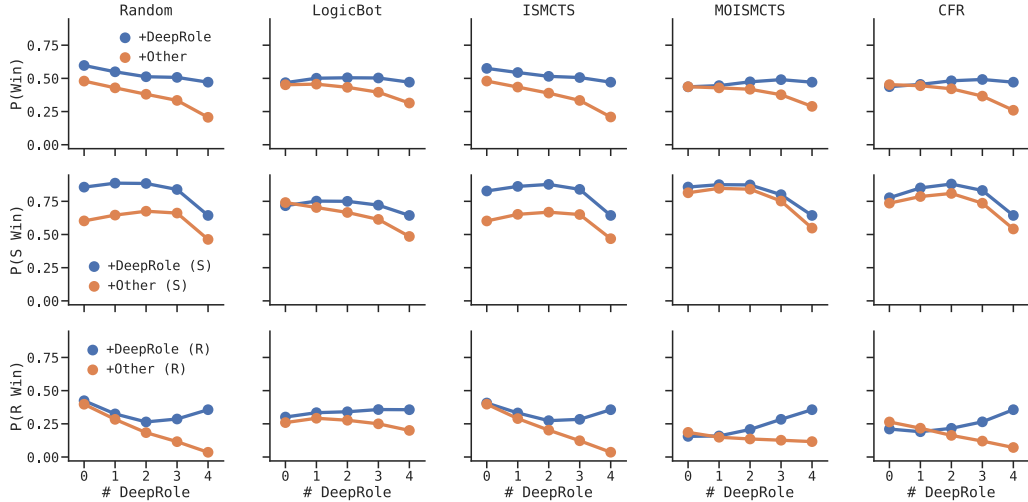

Figure 4: Comparing the expected win rate of DeepRole with other agents. The x-axis shows how many of the first four agents are DeepRole. The y-axis shows the expected win rate for the fifth agent if they played as DeepRole or the benchmark. Standard errors smaller than the size of the dots. (top) Combined expected win rate. (middle) Spy-only win rate. (bottom) Resistance-only win rate.

activation. This layer is designed to take into account the specific structure of $V$, respecting the binary nature of payoffs in Avalon (players can only win or lose). It explicitly represents the probability of a Resistance win ($\vec{\mathbf{w}} = P(\text{win}|\rho)$) for each assignment $\rho$.

Using these probabilities, we then calculate the $V_i(I)$ for each player and information set, constraining the network's outputs to sound values. To do this calculation, for each player $i$, win probabilities are first converted to expected values ($\vec{u}_i\vec{\mathbf{w}} + \text{-}\vec{u}_i(1 - \vec{\mathbf{w}})$) representing $i$'s payoff in each $\rho$ if resistance win. It is then turned into the probability-weighted value of each information set which is used and produced by CFR: $\vec{V}_i = M_i[(\vec{u}_i\vec{\mathbf{w}} + \text{-}\vec{u}_i(1 - \vec{\mathbf{w}})) \odot \mathbf{b}]$ where $M_i$ is a $(15 \times 60)$ multi-one-hot matrix mapping each $\rho$ to player $i$'s information set, and $\mathbf{b}$ is the belief over roles passed to the network. This architecture is fully differentiable and is trained via back-propagation. A diagram and description of the full network is shown in Figure 2. See Appendix B and Alg. 3 for details of the network training algorithm, procedure, parameters and compute details.

The win probability layer enabled training with less training data and better generalization. When compared to a lesioned neural network that replaced the win probability layer with a zero-sum layer (like DeepStack) the average held-out loss per network was higher and more training data was required (Figure 3).

## 4 Empirical game-theoretic analysis

The possibility of playing with diverse teammates who may be playing conflicting equilibrium strategies, out-of-equilibrium strategies, or even human strategies makes evaluation outside of two-player zero-sum games challenging. In two-player zero-sum games, all Nash equilibria are minimally exploitable, so algorithms that converge to Nash are provably optimal in that sense. However evaluating 3+ player interactions requires considering multiple equilibria and metrics that account for coordinating with teammates. Further, Elo and its variants such as TrueSkill are only good measures of performance when relative skill is intransitive, but have no predictive power in transitive games (e.g., rock-paper-scissors) [40]. Thus, we turn to methods for empirical game-theoretic analysis which require running agents against a wide variety of benchmark opponents [40, 42].

We compare the performance of DeepRole to 5 alternative baseline agents: CFR – an agent trained with MCCFR [22] over a hand-crafted game abstraction; LogicBot – a hand-crafted strategy that uses logical deduction; RandomBot - plays randomly; ISMCTS - a single-observer ISMCTS algorithm

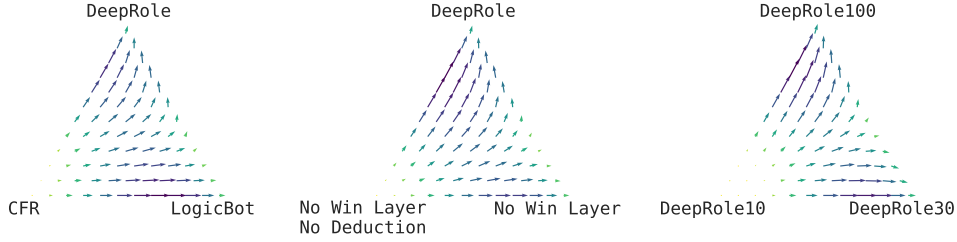

Figure 5: Empirical game-theoretic evaluation. Arrow size and darkness are proportional to the size of the gradient. (left) DeepRole against hand-coded agents. (center) DeepRole compared to systems without our algorithmic improvements. (right) DeepRole against itself but with CFR iterations equal to the number next to the game.

found in [11, 12, 35]; MOISMCTS - a multiple-observer variant of ISMCTS [43]. Details for these agents are found in Appendix C.

We first investigated the conditional win rates for each baseline agent playing against DeepRole. We consider the effect of adding a 5th agent to a preset group of agents and compare DeepRole's win rate as the 5th agent with the win rate of a baseline strategy as the 5th agent in that same preset group. For each preset group (0-4 DeepRole agents) we simulated >20K games.

Figure 4 shows the win probability of each of these bots when playing DeepRole both overall and when conditioning on the role (Spy or Resistance). In most cases adding a 5th DeepRole player yielded a higher win rate than adding any of the other bots. This was true in every case we tested when there were at least two other DeepRole agents playing. Thus from an evolutionary perspective, DeepRole is robust to invasion from all of these agent types and in almost all cases outperforms the baselines even when it is the minority.

To formalize these intuitions we construct a meta-game where players select a mixed meta-strategy over agent types rather than actions. Figure 5 shows the gradient of the replicator dynamic in these meta-games [40, 42]. The replicator dynamic gradient describes the direction a player playing meta-strategy $\sigma$ can update their strategy for maximal gain, assuming other players are also playing $\sigma$. Both vector field sinks and points with zero gradient correspond to Nash equilibria in the meta-game.

First, we compare DeepRole to the two hand-crafted strategies (LogicBot and CFR), and show that purely playing DeepRole is the equilibrium with the largest basin of attraction. The ISMCTS agents are too computationally demanding to run in these contests, but in a pairwise evaluation, playing DeepRole is the sole equilibrium.

Next, we test whether our innovations make DeepRole a stronger agent. We compare DeepRole to two lesioned alternatives. The first, DeepRole (No Win Layer), uses a zero-sum sum layer instead of our win probability layer in the neural network. Otherwise it is identical to DeepRole. In Figure 3, we saw that this neural network architecture did not generalize as well. We also compare to a version of DeepRole that does not include the logical deduction step in equation 1, and also uses the zero-sum layer instead of the probability win layer (No Win Layer, No Deduction). The agent without deduction is the weakest, and the full DeepRole agent is the strongest, showing that our innovations lead to enhanced performance.

Finally, we looked at the impact of CFR solving iterations during play (thinking time). More iterations make each move slower but may yield a better strategy. When playing DeepRole variants with 10, 30, and 100 iterations against each other, each variant was robust to invasion by the others but the more iterations used, the larger the basin of attraction (Figure 5).

## 5 Human evaluation

Playing with and against human players is a strong test of generalization. First, humans are likely to play a diverse set of strategies that will be challenging for DeepRole to respond to. During training time, it never learns from any human data and so its abilities to play with people must be the result of playing a strategy that generalizes to human play. Importantly, even if human players take the

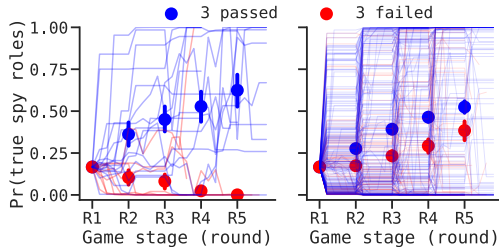

Figure 6: Belief dynamics over the course of the game. (left) DeepRole's posterior belief in the ground truth Spy role assignments as a Resistance player with four humans. (right) DeepRole's posterior belief of the true spy team while observing all-human games from the perspective a Resistance player.

DeepRole neural networks "out of distribution", the online CFR iterations can still enable smart play in novel situations (as with MCTS in AlphaGo).

Humans played with DeepRole on the popular online platform ProAvalon.com (see Appendix F for commentated games and brief descriptions of DeepRole's "play style"). In the 2189 mixed human/agent games we collected, all humans knew which players were human and which were DeepRole. There were no restrictions on chat usage for the human players, but DeepRole did not say anything and did not process sent messages. Table 1 shows the win rate of DeepRole compared to humans. On the left, we can see that DeepRole is robust; when four of the players were DeepRole, a player would do better playing the DeepRole strategy than playing as an average human, regardless of team. More interestingly, when considering a game of four humans, the humans were better off playing with the DeepRole agent as a teammate than another human, again regardless of team. Although we have no way quantifying the absolute skill level of these players, among this pool of avid Avalon players, DeepRole acted as both a superior cooperator and competitor – it cooperated with its teammates to compete against the others.

Finally, DeepRole's interpretable belief state can be used to gain insights into play. In Figure 6 we show DeepRole's posterior probability estimate of the true set of Spies when playing as a Resistance player. When DeepRole played as the sole agent among four humans (left plot), the belief state rapidly converged to the ground truth in the situations where three missions passed, even though it had never been trained on human data. If three missions failed, it was often because it failed to learn correctly. Next, we analyze the belief state when fed actions and observations from the perspective of a human resistance player playing against a group of humans (yoked actions). As shown in Figure 6, the belief estimates increase as the game progresses, indicating DeepRole can make correct inferences even while just observing the game. The belief estimate converges to the correct state faster in games with three passes, presumably because the data in these games was more informative to all players.

## 6  Discussion

We developed a new algorithm for multi-agent games called DeepRole which effectively collaborates and competes with a diverse set of agents in The Resistance: Avalon. DeepRole surpassed both humans and existing machines in both simulated contests against other agents and a real-world evaluation with human Avalon players. These results are achieved through the addition of a deductive reasoning system to vector-based CFR and a win probability layer in deep value networks for depth-

| | Adding DeepRole or a Human | | | | | | | |
| | to 4 DeepRole | | | | to 4 Human | | | |
| | +DeepRole | | +Human | | +DeepRole | | +Human | |
| | Win Rate (%) | (N) | Win Rate (%) | (N) | Win Rate (%) | (N) | Win Rate (%) | (N) |
|---|---|---|---|---|---|---|---|---|
| Overall | **46.9 ± 0.6** | (7500) | 38.8 ± 1.3 | (1451) | **60.0 ± 5.5** | (80) | 48.1 ± 1.2 | (1675) |
| Resistance | **34.4 ± 0.7** | (4500) | 25.6 ± 1.5 | (856) | **51.4 ± 8.2** | (37) | 40.3 ± 1.5 | (1005) |
| Spy | **65.6 ± 0.9** | (3000) | 57.8 ± 2.0 | (595) | **67.4 ± 7.1** | (43) | 59.7 ± 1.9 | (670) |

Table 1: Win rates for humans playing with and against the DeepRole agent. When a human replaces a DeepRole agent in a group of 5 DeepRole agents, the win rate goes down for the team that the human joins. When a DeepRole agent replaces a human in a group of 5 humans, the win rate goes up for the team the DeepRole agent joins. Averages ± standard errors of the mean calculated over a binary outcome.

limited search. Taken together, these innovations allow DeepRole to scale to the full game of Avalon allowing CFR agents to play hidden role games for the first time. In future work, we will investigate whether the interpretable belief state of DeepRole could also be used to ground language, enabling better coordination through communication.

Looking forward, hidden role games are an exciting opportunity for developing AI agents. They capture the ambiguous nature of day-to-day interactions with others and go beyond the strictly adversarial nature of two-player zero-sum games. Only by studying 3+ player environments can we start to capture some of the richness of human social interactions including alliances, relationships, teams, and friendships [32].

### Acknowledgments

We thank Victor Kuo and ProAvalon.com for help integrating DeepRole with human players online. We also thank Noam Brown, Murray Campbell, and Michael Wellman for helpful discussions and comments. This work was supported by Harvard Data Science Initiative, CRCS, and MBB, The Future of Life Institute, DARPA Ground Truth, and the Center for Brains, Minds and Machines (NSF STC award CCF-1231216).

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
