[Supplementary Material]

# A DeepRole depth-limited CFR

---

**Algorithm 1** DeepRole depth-limited CFR

---

1: **INPUT** $h$ (root public game history); $\mathbf{b}$ (root public belief); $n$ (# iterations); $d$ (averaging delay); NN[$h$] (neural networks that approximate CFVs from $h$)

   Init regrets $\forall I, r_I[a] \leftarrow 0$, Init cumulative strategies $\forall I, s_I[a] \leftarrow 0$

2: **procedure** SOLVESITUATION($h, \mathbf{b}, n, d$)
3:      $\vec{u}_{1\ldots p} \leftarrow \vec{0}$
4:      **for** $i = 1$ to $n$ **do**
5:          $w_i \leftarrow \max(i - d, 0)$
6:          $\vec{u}_{1\ldots p} \leftarrow \vec{u}_{1\ldots p} + $ MODIFIEDCFR+$(h, \mathbf{b}, w_i, \vec{1}_{1\ldots p})$
7:      **end for**
8:      **return** $\vec{u}_{1\ldots p} \,/\, \sum w_i$
9: **end procedure**

10: **procedure** MODIFIEDCFR+$(h, \mathbf{b}, w, \vec{\pi}_{1\ldots p})$
11:      **if** $h \in Z$ **then**
12:          **return** TERMINALCFVS$(h, \mathbf{b}, \vec{\pi}_{1\ldots p})$
13:      **end if**
14:      **if** $h \in$ NN **then**
15:          **return** NEURALCFVS$(h, \mathbf{b}, \vec{\pi}_{1\ldots p})$
16:      **end if**
17:      $\vec{u}_{1\ldots p} \leftarrow \vec{0}$
18:      **for** $i \in P'(h)$ **do**   ▷ A strategy must be calculated for all moving players at public history $h$
19:          $\vec{I}_i \leftarrow $ lookupInfosets$_i(h)$
20:          $\vec{\sigma}_i \leftarrow $ regretMatching+$(\vec{I}_i)$
21:      **end for**
22:      **for** public observation $o \in O(h)$ **do**
23:          $\vec{a}_{1\ldots p} \leftarrow $ deduceActions$(h, o)$
24:          **for** $i \in P'(h)$ **do**
25:              $\vec{\pi}_i \leftarrow \vec{\sigma}_i[\vec{a}_i] \odot \vec{\pi}_i$
26:          **end for**
27:          $\vec{u}'_{1\ldots p} \leftarrow $ MODIFIEDCFR+$(ho, \mathbf{b}, w, \vec{\pi}_{1\ldots p})$
28:          **for** each player $i$ **do**
29:              **if** $i \in P'(h)$ **then**
30:                  $\vec{m}_i[\vec{a}_i] \leftarrow \vec{m}_i[\vec{a}_i] + \vec{u}_i$
31:                  $\vec{u}_i \leftarrow \vec{u}_i + \vec{\sigma}_i[\vec{a}_i] \odot \vec{u}'_i$
32:              **else**
33:                  $\vec{u}_i \leftarrow \vec{u}_i + \vec{u}'_i$
34:              **end if**
35:          **end for**
36:      **end for**
37:      **for** $i \in P'(h)$ **do** ▷ Similar to line 18, we must perform these updates for all moving players
38:          **for** $I \in \vec{I}_i$ **do**
39:              **for** $a \in A(I)$ **do**
40:                  $r_I[a] \leftarrow \max(r_I[a] + \vec{m}_i[a][I] - \vec{u}_i[I], 0)$
41:                  $s_I[a] \leftarrow s_I[a] + \vec{\pi}_i[I]\vec{\sigma}_i[I][a]w$
42:              **end for**
43:          **end for**
44:      **end for**
45:      **return** $\vec{u}_{1\ldots p}$
46: **end procedure**

---

---

**Algorithm 2** Terminal value calculation

---

1: **procedure** TERMINALCFVS($h, \mathbf{b}, \vec{\pi}_{1...p}$)
2:      $\vec{v}_{1...p}[\cdot] \leftarrow 0$                     ▷ Initialize factual values
3:      $\mathbf{b}_{\text{term}} \leftarrow$ CALCTERMINALBELIEF($h, \mathbf{b}, \vec{\pi}_{1...p}$)
4:      **for** $i = 1$ to $p$ **do**
5:          **for** $\rho \in \mathbf{b}_{\text{term}}$ **do**
6:              $\vec{v}_i[I_i(h, \rho)] \leftarrow \vec{v}_i[I_i(h, \rho)] + \mathbf{b}_{\text{term}}[\rho]u_i(h, \rho)$
7:          **end for**
8:      **end for**
9:      **return** $\vec{v}_{1...p}/\vec{\pi}_{1...p}$            ▷ Convert factual to counterfactual
10: **end procedure**

11: **procedure** NEURALCFVS($h, \mathbf{b}, \vec{\pi}_{1...p}$)
12:      $\mathbf{b}_{\text{term}} \leftarrow$ CALCTERMINALBELIEF($h, \mathbf{b}, \vec{\pi}_{1...p}$)
13:      $w \leftarrow \sum_\rho \mathbf{b}_{\text{term}}[\rho]$
14:      $\vec{v}_1, \vec{v}_2, \ldots, \vec{v}_p \leftarrow$ NN$[h](h, \mathbf{b}_{term}/w)$      ▷ Call NN with normalized belief
15:      **return** $w\vec{v}_{1...p}/\vec{\pi}_{1...p}$         ▷ Convert factual to counterfactual
16: **end procedure**

17: **procedure** CALCTERMINALBELIEF($h, \mathbf{b}, \vec{\pi}_{1...p}$)
18:      **for** $\rho \in \mathbf{b}$ **do**
19:          $\mathbf{b}_{\text{term}}[\rho] \leftarrow \mathbf{b}[\rho] \prod_i \vec{\pi}_i(I_i(h, \rho))$
20:          $\mathbf{b}_{\text{term}}[\rho] \leftarrow \mathbf{b}_{\text{term}}[\rho](1 - \mathbb{1}\{h \vdash \neg\rho\})$      ▷ Zero beliefs that are logically inconsistent
21:      **end for**
22:      **return** $\mathbf{b}_{\text{term}}$
23: **end procedure**

---

# B   Value network training

We generate training data for the deep value networks by using CFR to solve each part of the game from a random sample of starting beliefs. By working backwards from the end of the game, trained networks from later stages enable data generation using CFR at progressively earlier stages. This progressive back-chaining follows the dependency graph of proposals shown on the left side of Figure 1. This generalizes the procedure used to generate DeepStack's value networks [27].

For each network, we sampled $120,000$ game situations ($\theta \in \Theta$) to be used for training and validation. For each sample, CFR ran for 1500 iterations, skipping the first 500 during averaging. The neural networks were each trained for 3000 epochs (batch size of 4096) using the Adam optimizer with a mean squared error loss on $\vec{V}$. Training hyperparameters and weight initializations used Keras defaults. 10% of the data was set aside for validation. Training on 480 CPU cores, 480 GB of memory, and 1 GPU took roughly 48 hours to produce the networks for every stage in the game.

# C   Comparison Agents

**CFR**   CFR denotes an agent using a strategy trained by external sampling MCCFR with a hand-built imperfect-recall abstraction, used to reduce the size of Avalon's immense game tree. We bucket information sets for players based on their initial information set (their role and who they see) and a set of hand-chosen game metrics: the round number, the number of failed missions each player has participated in, and the number of times a player has proposed a failing mission. We trained the bot until we observed decayed performance of the bot in self-play. In total, CFRBot was trained for 6,000,000 iterations.

**LogicBot**   LogicBot is an agent that plays a hand-crafted pre-set strategy derived from our intuition of playing Avalon with real people. During play, LogicBot keeps a list of possible role assignments that are logically consistent with the observations it has made. As resistance, it randomly samples an assignment and proposes a mission using the resistance players in that assignment. It votes up proposals if and only if the proposed players and the proposer are resistance in a randomly sampled assignment or if it is the final proposal in the round. As spy, it proposes randomly, votes opposite to resistance players, and selects merlin randomly.

**Random**   Our random agent selects an action uniformly from the available actions.

**Algorithm 3** Backwards training

1: **INPUT** $P_{1...n}$: Dependency-ordered list of game parts.
2: **INPUT** $\Theta_{1...n}$: For each game part, a distribution over game situations.
3: **INPUT** $d$: The number of training datapoints generated per game partition.
4: **OUTPUT** $N_{1...n}$: $n$ trained neural value networks, one for each game part.

5: **procedure** ENDTOENDTRAIN($P_{1...n}, \Theta_{1...n}, d$)          ▷ Train a neural network for each game partition
6:     **for** $i = 1$ to $n$ **do**
7:         $\mathbf{x}, \mathbf{y} \leftarrow$ GENERATEDATAPOINTS($P_i, \Theta_i, N_{1...i-1}$)
8:         $N_i \leftarrow$ TRAINNN($\mathbf{x}, \mathbf{y}$)
9:     **end for**
10:     **return** $N_{1...n}$
11: **end procedure**

12: **procedure** GENERATEDATAPOINTS($d, S, \Theta, N_{1...k}$) ▷ Given a game partition, it's distribution over game situations, and the NNs needed to limit solution depth, generate $d$ datapoints.
13:     **for** $i = 1$ to $d$ **do**
14:         $\theta_i \sim \Theta$                          ▷ Sample a game situation from the distribution
15:         $\mathbf{v}_i \leftarrow$ SOLVESITUATION($S, \theta_i, N_{1...k}$)     ▷ Solve that game situation for every player's values, using previously trained neural networks to solution depth.
16:     **end for**
17:     **return** $\theta_{1...d}, \mathbf{v}_{1...d}$                          ▷ Return all training datapoints
18: **end procedure**

---

**Algorithm 4** Game Situation Sampler

1: **INPUT** $s$: The number of succeeds.
2: **INPUT** $f$: The number of fails.
3: **OUTPUT** $p, \mathbf{b}$: A random game situation from this game part, consisting of a proposer and a belief over the roles.

4: **procedure** SAMPLESITUATION($s, f$)
5:     $I \leftarrow$ SAMPLEFAILEDMISSIONS($s, f$)                          ▷ Uniformly sample $f$ failed missions
6:     $E \leftarrow$ EVILPLAYERS($I$)                    ▷ Calculate evil teams consistent with the missions
7:     $P(E) \sim \text{Dir}(\vec{1}_{|E|})$                                  ▷ Sample probability of each evil team
8:     $P(M) \sim \text{Dir}(\vec{1}_n)$                    ▷ Sample probability of being Merlin for all players
9:     $\mathbf{b} \leftarrow P(E) \bigotimes P(M)$             ▷ Create a belief distribution using $P(E)$ and $P(M)$
10:     $p \sim \text{unif}\{1, n\}$                    ▷ Sample a proposer uniformly over all the players
11:     **return** $p, \mathbf{b}$
12: **end procedure**

Figure 7: An example of the regrets stored at each node in the game tree. During the search phase, the ModifiedCFR+ algorithm updates these regrets iteratively on a shortened version of the full game tree, using NeuralCFVs as the leaf evaluators.

**ISMCTS & MOISMCTS**    We also evaluate our bot against opponents playing using the ISMCTS family of algorithms. Specifically, we evaluate our bot against the single-observer ISMCTS (ISMCTS) algorithm shown in [11, 12, 43], as well as the improved multiple-observer version of the algorithm (MOISMCTS). Each variant used 10,000 iterations per move.

## D    State space calculation

Unlike large two-player games like Poker, Go, or Chess, Avalon's complexity lies in the combinatorial explosion that comes with having 5 players, four role types (Spy, Resistance, Merlin, Assassin), and a large number observable moves. We lower bound the number of information sets by just considering the longest possible game. The longest possible game lasts five rounds with each round requiring five proposals. Each proposal can made in 10 different ways by choosing which 2 or 3 players out of 5 should go on the mission. From there, there are 16 ways proposals 1-4 can be voted down and 16 ways proposal 5 can be voted up. Thus, a lower bound on the number of information sets is $(10 * 16)^{5*5} \approx 10^{56}$ which does not consider shorter games or any of the private information.

## E    ProAvalon.com

`ProAvalon.com` is a website where players can play Avalon online in groups of 5 to 10. We've integrated DeepRole in to this website, allowing humans from all around the world to play against 0-4 DeepRole agents. Fig. 8 shows the game interface for human players on ProAvalon.com. Natural language communication is done via a publicly visible chat. See the website for more details about the specific interface.

## F    Human commentary of DeepRole v. Human games.

Some players on ProAvalon.com have uploaded commentary that qualitatively examine the style of play the bots have. We examine two of these games to show DeepRole effectively cooperating and competing with a human player.

In the first game we examine (`https://www.youtube.com/watch?v=LKdY4Us0Ci4`), the human player is playing as "VT" ("Vanilla Town", i.e. non-Merlin resistance). After the first two missions fail and the third one passes, the human player is able to accurately deduce the identities of the spy players. During proposals for the

4th and 5th missions, however, his fellow resistance teammates (including Merlin), seem to be rejecting missions that he knows to be "clean" (do not contain a spy). While he expresses exasperation that one of his teammates doesn't seem to deduce the obvious, these clean missions are eventually approved and succeed. At the end of the game, resistance win, revealing that the rejecting player was Merlin all along – purposefully rejecting missions to seem ignorant.

In the second game we examine (`https://www.youtube.com/watch?v=9RkUFHYTo_s`), the human player is playing as Merlin. During multiple rounds of the game, the human player "slams clean", proposing a mission containing no spies – generally an obvious indicator of Merlin-like knowledge of the spy players. While these missions are ultimately approved and succeeded, the DeepRole Assassin correctly identifies the human player due to their obvious play, resulting in a spy victory.

There are more examples of DeepRole v. human games on YouTube, and we encourage readers to check out other videos with qualitative analysis of DeepRole.

Figure 8: The ProAvalon.com game interface. This shows a completed game between 4 DeepRole agents and a human player (no affiliation to this work's authors). The interface consists of a visualization of a "round table" of players (top), a public chat for each game (bottom left), and the public game history (bottom right).