[Reviews · NeurIPS 2019]

Reviewer 1



Originality: To the best of my knowledge, this is the first method that surpasses human performance in a multi-player game where actions can be hidden during the game. Most of the components are extensions of DeepStack [1], but new formalizations (e.g. deductive reasoning for inferring the private actions) and model architecture adjustments (e.g. win probability layer) were necessary for addressing the challenges of the game under consideration. Quality & Clarity: Both the experimental setup and the proposed algorithm are clearly presented. The paper is well written and self-contained for the most of it. Significance: The extension of CFR to hidden action games can be very impactful, and the steps to achieve this were non-trivial!

Reviewer 2



Originality is sufficient. They proposed a novel combination of existing methods: counterfactual regret minimization, value network trained during self-play, and reasoning. The quality of this paper is good. The authors are knowledgeable about this area. The evaluation; discussion is careful and insightful. The paper is clearly written.

Reviewer 3



The paper builds on well-known methods (CFR) and provides novel improvements and modifications that extend the approach to a multiplayer, hidden-role setting. This is original and novel and creative, though the crucial role of CFR cannot be understated. Related work appears to be adequately cited. The empirical results provide the main validation for the soundness and quality of the proposed algorithm; this is reasonable and is explained well in the paper. I have not spotted any obvious illogicalities or mistakes. The paper is mostly well-written and logically organized, and I think that with the supplement it would be possible to reproduce the results with some effort. Moreover, as part of their response the authors have made their code and data publicly available. The authors have also provided satisfactory responses to the following issues raised in the first version of this review: There is some unexplained notation: sigma_{I->a}^t on line 117, z[I] on line 118. Figure 5 and line 230 talk about the gradient of the "replicator dynamic," which is not explained in the text; while the figure is intuitively easy to interpret, its precise meaning is not adequately explained in the paper. On a higher level, I would liked to see a brief explanation of CFR(+) as I was not very familiar with its workings, but I understand that there is limited space available and the approach is well explained in the cited literature. I think the results are important both methodologically and empirically. Multiplayer games with hidden information present a very difficult field and this work presents significant advancements with very good empirical results. While the architecture does have some inherent scalability issues (due to the large number of distinct neural networks needed), this can possibly be addressed in future work, and anyhow the authors obtain their results with surprisingly small networks. I believe that this work is directly applicable to certain other games and scenarios and that this will be a good foundation for future research.

[Author Response · NeurIPS 2019]

We thank all three reviewers for their comments and insightful suggestions. We have edited the manuscript to address
them. We outline some of these changes here. One important addition is that we have now packaged our source code
with trained model weights and data for open source release [1]. We believe that the availability of this code will aid in
the reproduction and extension of our results by other researchers in the community.

*I think the author needs to argue why Avalon is a better agent for real-world hidden role scenarios than other*
*games?* Among hidden role games, Avalon is one of the most popular and widely played games (according to
boardgamegeek.com) including an active online community of player (proavalon.com). In the real world, there are
subtle cues which can often be misinterpreted when others are acting under uncertainty. Avalon is not necessarily
fundamentally better than other games like Saboteur, but its combination of discrete actions and natural language as
well as its active online community make it a good candidate for study.

*For the methodology part, can you differentiate your method from existing methods such as MCTS + value network*
*in alphaGO more explicitly?* Our approach uses CFR instead of MCTS. AlphaGo-like methods can be used when
the board state alone is sufficient to determine the best move, but in imperfect information games it is necessary to
consider how players acted to reach the current board state. We've added the following sentence: "Compared to
MCTS-based methods like AlphaGo, CFR-based methods like DeepStack and DeepRole can soundly reason over
hidden information."

*Does the proposed method generalize to other games such as werewolf or saboteur? . . . Do we actually want to a*
*case-by-case AI or general intelligent agents?* We agree that the goal is to create generally intelligent agents rather
than specific agents designed to play a particular game. The DeepRole algorithm extends the CFR algorithm which
was developed for two-player competitive imperfect information games such as poker. We showed that CFR and value
networks can be used to build a more general system capable of cooperation in addition to competition. In principle,
DeepRole could be applied directly to Saboteur. However, for other hidden role games such as Werewolf or Mafia,
understanding and producing natural language is a key missing component. We mention in the discussion: "In future
work, we will investigate whether the interpretable belief state of DeepRole could also be used to ground language,
enabling better coordination through communication."

*Need ablation and analysis — we all know trained agents are vulnerable to adversarial human players — e.g. the*
*online dota bots, who beat professionals, can be easily beaten by streamers after a few days. Are you claiming your*
*bot is not 'hackable'?* We have carried out experiments with lesioned versions of DeepRole which showed that our
novel neural network architecture and deductive reasoning component were key drivers of performance (Figure 5). We
are not claiming that the bot is not exploitable. Because it is possible for agents to team up on others in flexible ways,
exploitability is less straightforward to measure. Additionally, over the 1-2 week period of online play, the Avalon
community (including streamers) did not collectively find a strategy that could consistently win against DeepRole.

*Another interesting observation is the bot does not need conversation. Does this mean the game is not well-designed or*
*a good strategy is to close your eyes and shut your mouth during playing?* We aren't sure, but this is an interesting
question. Algorithms like DeepRole could be used to to probe these ideas and challenge human players to find a distinct
role for language.

*Clarify unexplained notation and terminology.* We've added the following sentence to
line 110: "...be the joint strategy of all $p$ players. We write $\sigma_{I \to a}$ to mean strategy $\sigma$,
modified so action $a$ is always played at information set $I$.". We've added the following
to the end of line 118: "..., where $z[I]$ is the $h \in I$ such that $h \sqsubseteq z$". We have also
added additional text on the "replicator dynamics" used for evaluation: "The replicator
dynamic gradient describes the direction a player playing meta-strategy $\sigma$ can update
their strategy for maximal gain, assuming other players are also playing $\sigma$. Both vector
field sinks and points with zero gradient correspond to nash equilibria in the replicator
dynamic gradient."

*On a higher level, I would liked to see a brief explanation of CFR(+). . .* We have
enriched the supplement to more clearly point to the full algorithm and have added
an additional figure which describes the complete architecture with all components in
place.

*Provide the raw data for all tables and figures for reproducibility.* This data is included in the open source repository.

*Also, Table 1 does not explain what the +/- indicates, nor whether e.g. approximate Gaussianity holds.* We have
changed the figure caption to read: "Confidence interval is the standard error of the mean calculated over a binary
outcome". This does not require the assumption of Gaussianity.

## Footnotes

[1]https://bitbucket.org/Anonymous314159/deeprole/src/master/


[Meta-Review · NeurIPS 2019]

All reviewers agree that the paper provides some nice contributions (extending CFR beyond 2 players and tackling Avalon) and that the authors succeed well with their rebuttal to address some of the major concerns brought on by some of the referees. They have responded adequately and furthermore open-sourced their implementation. We expect the authors though to carry out the promised changes (and also improve on the notation). Overall a good solid paper.